# ExpAct: Building and Evolving Structural and Procedural Experience for Mobile GUI Agents

## Abstract

While Graphical User Interface (GUI) agents have shown promise in automating mobile interactions, most existing methods emphasize zero-shot execution or task generalization, and largely overlook the continual accumulation, organization, and reuse of interaction experience over time. We introduce **ExpAct**, an experience-centric framework that builds and progressively evolves structural and procedural long-term experience from historical interaction trajectories. ExpAct operates in three stages: (1) Stage 1 abstracts raw execution traces into widget-centric relational representations capturing interface structure, and subtask-conditioned procedural patterns encoding reusable path strategies. (2) Stage 2 evolves long-term experience through an experience completion and refinement mechanism. (3) Stage 3 incorporates a subtask-wise retrieval mechanism to index and recall relevant experiences, providing execution-time guidance. Experiments on AndroidWorld show that ExpAct significantly outperforms zero-shot and naive episodic textual baselines in both task success rate and efficiency. We further introduce AndroidWorld-LTB, a long-tail benchmark derived from AndroidWorld that targets long-tail tasks with low-frequency widgets and sparse interaction patterns, where ExpAct demonstrates robust performance advantages.

## 1. Introduction

Recent advances in graphical user interface (GUI) agents (Zhang et al., 2023; Wang et al., 2024; Ye et al., 2025; Li et al., 2025a; Liu et al., 2024; Wang et al., 2025; Agashe et al., 2025) have demonstrated promising capabilities in

[1]Anonymous Institution, Anonymous City, Anonymous Region, Anonymous Country. Correspondence to: Anonymous Author <anon.email@domain.com>.

Preliminary work. Under review by the International Conference on Machine Learning (ICML). Do not distribute.

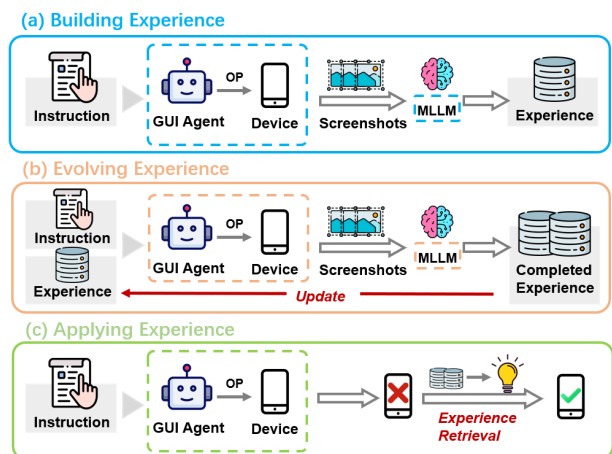

*Figure 1.* ExpAct comprises three main functionalities. (a) Constructing structural and procedural experience from historical interaction trajectories. (b) Iteratively completing and refining experience through targeted re-execution. (c) Retrieving subtask-aligned experience during execution to guide the agent and avoid repeating known failures, without modifying the underlying agent.

automating complex interactions on mobile applications, driven by progress in multimodal large language models (MLLMs) (Qin et al., 2025; Ye et al., 2025; Yang et al., 2025). Modern GUI agents typically leverage MLLMs to jointly process visual observations such as screenshots containing UI layouts and widgets together with textual task instructions, enabling unified reasoning over interface perception, task understanding, and action generation. Concretely, given a screen state, the MLLM interprets UI elements, infers their semantic roles and affordances, and selects the next action, such as tapping, scrolling, or text input, often through intermediate planning or chain-of-thought reasoning. These capabilities allow agents to plan long-horizon action sequences and operate over diverse application interfaces without task-specific engineering. However, in most existing systems, execution is treated as episodic. While short-term context is maintained within a single task execution, agents fail to accumulate or reuse interaction experience across tasks, even when tasks share similar goals, interface layouts, or recurring interaction patterns (Yao et al., 2022). Consequently, agents repeatedly re-interpret familiar screens from scratch, re-explore previously visited inter-

face regions, and re-commit similar mistakes, leading to limited efficiency, suboptimal path selection, and reduced robustness when interaction structures recur across tasks.

Recent work has explored knowledge retrieval–augmented methods for agents to address this limitation (Lewis et al., 2020; Zhang et al., 2025a), while most approaches still rely on external knowledge sources (Li et al., 2025b) or pre-constructed knowledge bases (Guan et al., 2025; Xie et al., 2025), which are often weakly grounded in the agent's own interaction dynamics and incur substantial construction and maintenance costs. In contrast, an agent's historical interaction trajectories constitute a rich but underutilized source of experience, capturing both procedural patterns and interface-specific regularities. This gap exposes two fundamental challenges in GUI agent design. *First*, interaction experience must be accumulated and organized in a reusable and scalable form as executions grow over time. *Second*, such experience must be retrieved and applied effectively during execution. GUI automation requires state- and subtask-dependent experience guidance rather than static factual recall, calling for retrieval mechanisms that align past experience with execution progress.

To address these challenges, we propose ExpAct for systematically accumulating and reusing interaction experience in GUI agents. ExpAct operates entirely at the experience layer, without modifying the underlying agent or learning new policies, and treats historical interaction trajectories as a form of internal, self-generated experience that can be incrementally consolidated and reused. Concretely, ExpAct integrates three tightly coupled components into a unified pipeline. First, it performs **composite experience consolidation (Stage 1)**, constructing complementary UI widget-level structural experience and subtask-level procedural experience from past executions, where structural experience characterizes reachable interface states, and procedural experience specifying how to proceed efficiently toward task completion. It then applies **experience completion and refinement(Stage 2)** to identify underrepresented or failure-prone interaction regions based on execution feedback, enabling targeted enrichment of experience coverage. Finally, ExpAct supports **execution-aware experience recall (Stage 3)**, retrieving relevant experience in a subtask-wise manner to guide decision-making during task execution.

To assess the effectiveness of structured experience reuse, we evaluate ExpAct on AndroidWorld (Rawles et al., 2024) and observe consistent improvements in task success rate and execution efficiency over baseline approaches. Moreover, we introduce **AndroidWorld-LTB**, a long-tail benchmark derived from AndroidWorld that focuses on tasks involving low-frequency widgets and sparse interaction patterns, referring to rarely encountered UI elements and infre-

quently executed but important subtask paths. This benchmark enables systematic evaluation of agent performance under long-tail and low-coverage conditions. In summary, our contributions are as follows:

- **Experience-Centric Perspective:** We reformulate GUI agent execution from an experience-centric viewpoint, identifying structured accumulation and reuse of interaction experience as a core system-level challenge.

- **ExpAct Experience Framework:** We propose ExpAct, an experience-centric framework that builds and progressively evolves structural and procedural long-term experience from historical interaction trajectories, as shown in Fig. 1, enabling subtask-aligned experience recall without modifying the underlying agent.

- **Long-Tail Benchmark:** We introduce AndroidWorld-LTB, a benchmark designed to evaluate agent performance on long-tail GUI tasks with low-frequency widgets and sparse interaction patterns, assessing robustness under sparse experience coverage.

## 2. Related Work

Multimodal large language models (MLLMs) have significantly enhanced the capabilities of mobile GUI agents (Zhang et al., 2023; Wang et al., 2024; Ye et al., 2025; Li et al., 2025a; Liu et al., 2024; Gemini Team, Google, 2023; Zhang et al., 2025b; Xiao et al., 2025). By integrating visual, textual, and structural information, MLLMs enable agents to reason over complex interfaces, plan long-horizon action sequences, and execute diverse tasks directly on screenshots or structured UI representations (Qin et al., 2025; Ye et al., 2025). Building on these capabilities, prior work has explored both model-centric and framework-based GUI agents, including model-based approaches such as UI-TARS (Qin et al., 2025) and Qwen2.5-VL (Ye et al., 2025), as well as system-level agents and frameworks such as AppAgent (Zhang et al., 2023), and Mobile-Agent-v3 (Ye et al., 2025). In parallel, the community has developed dedicated benchmarks that provide standardized evaluation environments for mobile GUI agents, facilitating systematic comparison and accelerating ecosystem development. Representative benchmarks include AndroidWorld (Rawles et al., 2024), AndroidLab (Xu et al., 2025), and other commonly used benchmarks (Chen et al., 2024; Chai et al., 2025; Kong et al., 2025).

Recent work has explored knowledge retrieval–augmented methods to enhance GUI agents' decision-making (Li et al., 2025b; Guan et al., 2025; Zhao et al., 2025; Xie et al., 2025). Some approaches, such as GUI-Explorer, improve the agent's knowledge application by incorporating autonomous exploration to acquire task-specific knowledge (Xie et al.,

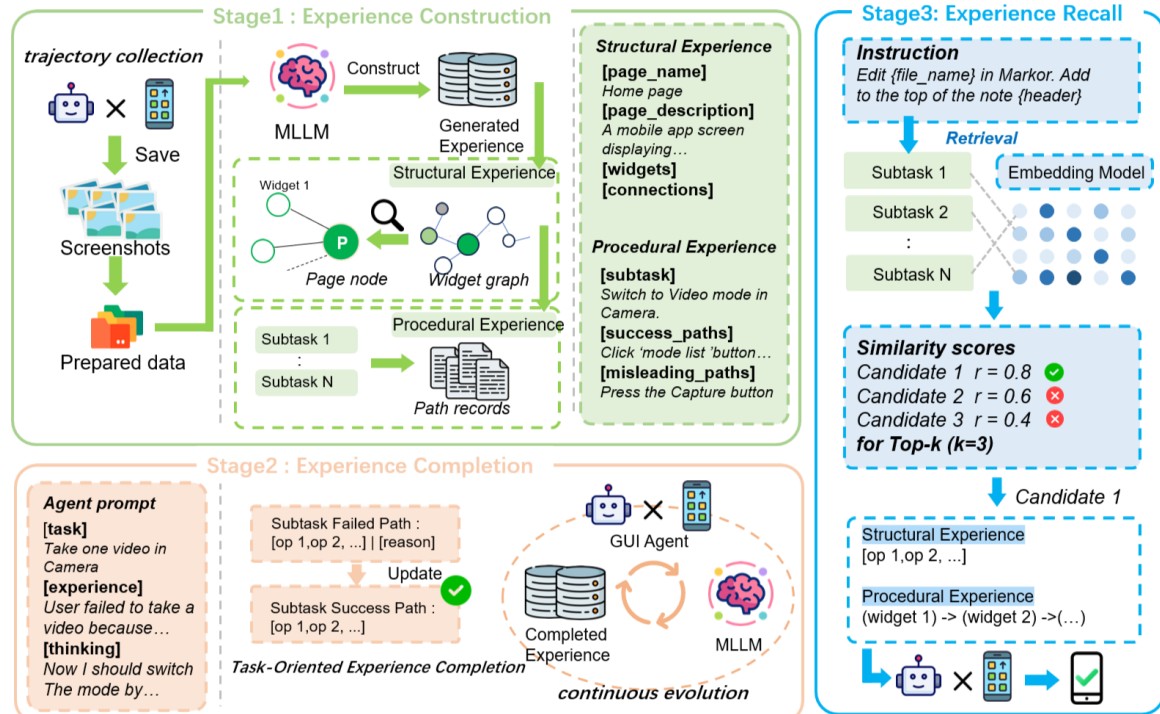

*Figure 2.* Overview of the ExpAct framework. ExpAct incrementally builds and evolves long-term experience from historical GUI interaction trajectories. Stage 1 constructs structural widget graphs and subtask-level procedural patterns from raw executions. Stage 2 refines and expands experience through iterative completion over failed and underexplored interactions. Stage 3 retrieves subtask-aligned experience at execution time to guide the agent and prevent repeated failures.

2025). Other methods, such as EchoTrail, focus on integrating external knowledge sources and refining them for downstream reasoning (Li et al., 2025b). While these approaches recognize the value of accumulated knowledge, they typically rely on large-scale exploration or pre-constructed knowledge bases, which makes it difficult to dynamically maintain and update the knowledge over time and incurs substantial computational and maintenance costs. Moreover, although KG-RAG (Guan et al., 2025) has explored interface structure using UTG graphs (Wen et al., 2024), the stored experience is ultimately represented at a coarse granularity, for example as task-to-action-path mappings, without fine-grained modeling of UI widget-level interaction paths or dynamic alignment with interface structure. More importantly, an agent's own historical interaction experience constitutes a rich yet underutilized source of procedural and interface-specific knowledge, offering the potential for incremental consolidation, targeted retrieval, and systematic reuse to support more reliable and efficient execution.

## 3. Method

In this section, we present **ExpAct**, a framework for constructing and evolving structural and procedural experience to enhance mobile GUI agents, following a three-stage strategy, as illustrated in Fig. 2.

### 3.1. Problem Formulation

We consider a GUI agent interacting with applications drawn from a stable environment distribution. At each time step $t$, the agent observes a GUI state $s_t \in \mathcal{S}$ and executes an action $a_t \in \mathcal{A}$, leading to a state transition $s_{t+1}$. A task is specified by a natural language instruction $\tau \in \mathcal{T}$, and an execution episode produces a trajectory $\pi = \{(s_1, a_1), (s_2, a_2), \dots\}$.

Let $\mathcal{A}_\theta$ be a base agent with fixed parameters $\theta$. ExpAct introduces an external experience store $\mathcal{E}_t = \{\mathcal{G}_t, \mathcal{P}_t\}$, where $\mathcal{G}_t$ is structural experience and $\mathcal{P}_t$ is procedural experience. The agent interacts with the environment and updates experience according to:

$$\begin{cases} s_{t+1} \sim \mathcal{T}_{\text{env}}(s_t, a_t) \\ a_t \sim \pi_\theta(s_t, \tau, \mathcal{E}_t) \\ \mathcal{E}_{t+1} = \mathcal{U}(\mathcal{E}_t, \pi) \end{cases} \quad (1)$$

where $\mathcal{T}_{\text{env}}$ is the environment transition, $\pi_\theta$ is the experience-augmented policy, and $\mathcal{U}$ updates experience with the latest trajectory $\pi$. The experience is progressively refined and evolved over multiple episodes, allowing agent to maximize task performance.

## 3.2. Stage I: Experience Construction

### 3.2.1. INTERFACE ABSTRACTION

To extract reusable and exploitable experience knowledge from high-dimensional raw trajectories of a GUI agent, we associate each GUI state $s_t$ with a corresponding screenshot and abstract it into a structured interface representation. Specifically, a state $s_t$ is denoted as

$$\begin{cases} \hat{s}_t \approx (\mathcal{W}_t, \mathcal{F}_t), \\ \mathcal{W}_t = \{w_t^1, \ldots, w_t^{n_t}\}, \end{cases} \tag{2}$$

where $\mathcal{W}_t$ is the set of UI widgets and $\mathcal{F}_t$ represents the page-level function constituted by these widgets. This abstraction serves as the foundation for ExpAct to construct both structural and procedural experience.

### 3.2.2. CONSTRUCTION FOR STRUCTURAL EXPERIENCE

ExpAct consolidates raw interaction trajectories into a directed widget graph $\mathcal{G} = (\mathcal{V}, \mathcal{E})$, where $\mathcal{V}$ denotes the set of GUI pages and $\mathcal{E}$ denotes navigational transitions between pages triggered by specific widgets, representing structural experience based on interface abstractions of GUI states. Specifically, an MLLM traverses each state in a task trajectory, extracting for each current GUI state $s_{t+1}$ its contained widgets, semantic descriptors (page descriptors), and triggering widget-action pairs. Each page node thus encodes both its internal structure and incoming/outgoing connections.

The MLLM evaluates whether the current state corresponds to the previous page, updating the existing node if a match is found. If no direct match exists, it retrieves a small set of candidate nodes from the existing graph based on structural and semantic embedding similarity, augmented with local connectivity cues, and applies a similarity threshold. In cases where multiple candidates exhibit comparable similarity, the MLLM serves as a semantic adjudicator, jointly reasoning over page descriptors and transition context to determine whether to merge the current representation $\hat{v}$ into an existing node or create a new one. This incremental consolidation allows ExpAct to preserve both executed transitions and plausible but unrealized interactions, resulting in a rich and reusable structural experience.

### 3.2.3. CONSTRUCTION FOR PROCEDURAL EXPERIENCE

ExpAct constructs procedural experience from historical trajectories at the subtask level, integrating textual path execution information. Each task is first decomposed by an LLM into a sequence of subtasks, and the trajectory is segmented accordingly. Within each subtask segment, exploratory, recovery, and redundant actions are removed to obtain a goal-driven action sequence, which is annotated with success or failure outcomes and the corresponding

failure causes. The extracted subtask experiences are then parameterized and templated to enable generalization across different instances. Procedural patterns are consolidated based on subtask semantics and action structures: successful experiences are prioritized to guide future executions, while failed experiences are retained as negative constraints. By leveraging subtask decomposition, trajectory analysis, and pattern abstraction, ExpAct captures both the success paths and misleading paths of each subtask, along with failure reasons, thereby constructing reusable operational patterns that inform and constrain subsequent executions.

### 3.2.4. UNIFIED EXPERIENCE REPRESENTATION

Taken together, the experience maintained by ExpAct can be formalized as a unified composition of structural and procedural components. Specifically, given a set of task trajectories $\mathcal{T}$, the overall experience repository $\mathcal{E}_{\text{ExpAct}}$ is defined as

$$\mathcal{E}_{\text{ExpAct}}(\mathcal{T}) = \left( \mathcal{G}(\mathcal{V}, \mathcal{E}, \Phi_s, \Phi_w), \bigcup_{g \in \mathcal{S}} \{(g, \tau_g, y_g, r_g)\} \right), \tag{3}$$

where the structural experience is represented by a directed widget graph $\mathcal{G}$ with page nodes $\mathcal{V}$, widget-triggered transitions $\mathcal{E}$, and associated structural and semantic representations $\Phi_s$ and $\Phi_w$, while the procedural experience consists of a collection of subtask-level execution patterns. For each subtask $g \in \mathcal{S}$, the procedural component stores its goal-driven action sequence $\tau_g$, execution outcome $y_g \in \{0, 1\}$, and failure rationale $r_g$ when applicable. This unified formulation highlights how ExpAct jointly captures interface-level structural regularities and subtask-level operational patterns, enabling experience to be incrementally accumulated, refined, and retrieved in an execution-aware manner.

## 3.3. Stage II: Experience Completion and Refinement

Experience accumulated from initial execution is inherently limited by one-pass exploration and execution noise. In particular, failed subtasks are often left uncorrected, misleading action sequences may be prematurely consolidated, and rare interactions or low-frequency interface regions remain insufficiently explored. As a result, the resulting experience exhibits systematic gaps and biases that cannot be resolved through passive accumulation alone. To address these issues, ExpAct introduces an iterative process of *experience completion and refinement*, which aims to (i) revisit and expand experience over failure cases and underexplored interactions, and (ii) progressively improve the fidelity and reliability of both structural and procedural experience by correcting incomplete or erroneous abstractions.

Concretely, ExpAct identifies failed tasks and re-executes them for additional rounds. During re-execution, the agent

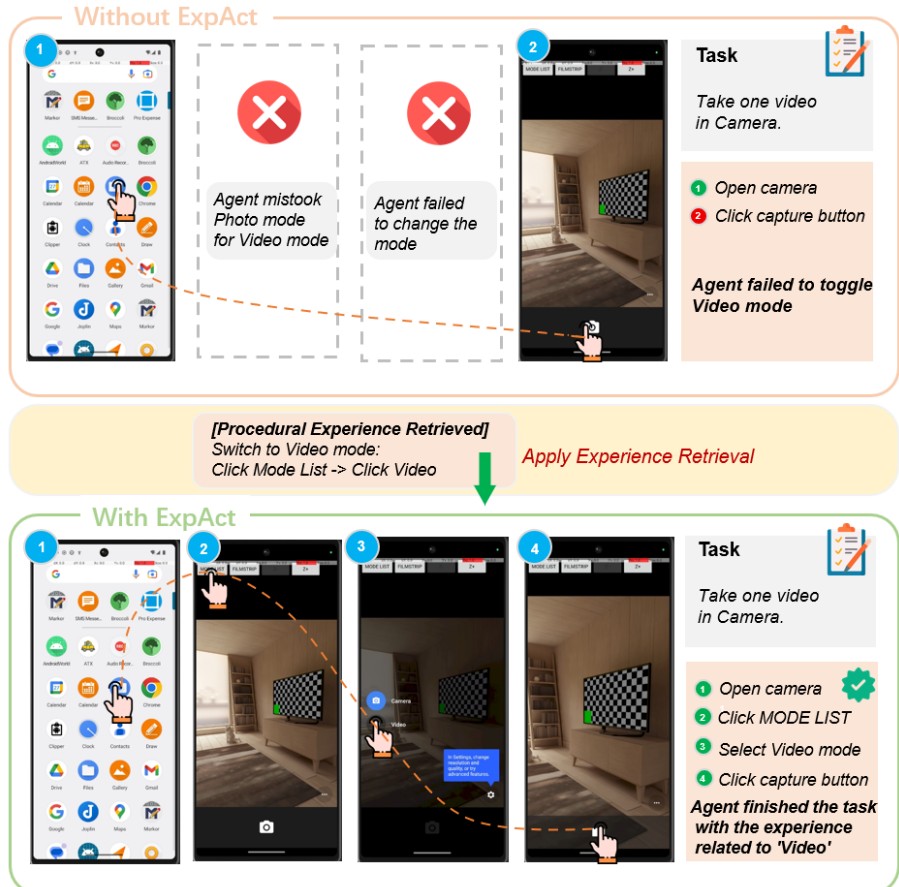

*Figure 3.* Case study of video recording in the Camera app. Without ExpAct (top), the agent directly triggers capture without switching modes, resulting in incorrect photo capture. With ExpAct (bottom), the agent recalls relevant experience, switches to Video mode, and successfully completes the task, demonstrating improved execution reliability through experience-augmented guidance.

is augmented with previously extracted procedural experience, where successful subtask trajectories serve as positive guidance and misleading paths act as negative constraints, while execution itself remains free of explicit subtask-level planning. This design allows the agent to avoid known failure modes while exploring alternative behaviors. After each re-execution, the resulting trajectory is re-segmented into subtasks, and newly observed outcomes are re-labeled to produce updated successful and failed subtask experiences. These refined procedural patterns are merged back into the experience repository, which can be formally expressed as an iterative update:

$$\mathcal{E}_{\text{ExpAct}}^{(k+1)} = \mathcal{E}_{\text{ExpAct}}^{(k)} \oplus \left( \Delta\mathcal{G}^{(k)}, \ \Delta\mathcal{P}^{(k)} \right), \qquad (4)$$

where $\mathcal{E}_{\text{ExpAct}}^{(k)}$ denotes the experience repository after the $k$-th iteration, $\Delta\mathcal{G}^{(k)}$ represents the newly observed or corrected structural experience (widget graph updates), $\Delta\mathcal{P}^{(k)}$ represents the updated procedural experience (subtask trajectories and success/failure annotations), and $\oplus$ denotes the incremental accumulation and merging of both structural and procedural components. Meanwhile, widget-level

structural experience is naturally enriched through repeated executions, as additional widgets, transitions, and sparsely activated interface regions are encountered and consolidated, resulting in a more balanced and information-rich experience representation.

### 3.4. Stage III: Execution-Aware Experience Retrieval

During task execution, ExpAct performs subtask-wise experience retrieval to provide the agent with dynamic, context-sensitive guidance. For the current task, the original instruction is first decomposed by an LLM into an ordered set of subtasks, and the currently active subtask is inferred to form a subtask-level query. Both the query and the retrieved experience are continuously updated based on the agent's progress within the current subtask. Upon completion of a subtask, the active subtask is updated, and the corresponding experience is refreshed accordingly. The refined subtask queries are then injected back into the agent's inference process as an experience-aware prompt, enabling the agent to reason over enriched, execution-relevant guidance rather than relying solely on the raw instruction. Experience in Ex-

*Table 1.* Performance comparison under experience-augmented execution on AndroidWorld.

| Method | Agent | | | |
|---|---|---|---|---|
| | UI-TARS-1.5-7B | | Qwen2.5-VL-7B | |
| | SR(%) ↑ | Avg. Steps ↓ | SR(%) ↑ | Avg. Steps ↓ |
| *Zero-shot* | | | | |
| No Experience | 35.3 | 22.8 | 28.4 | 25.1 |
| *Naive Episodic Textual Methods* | | | | |
| Raw Trajectory Experience | 38.8 | 21.6 | 31.9 | 23.9 |
| Abstracted Text Experience | 43.1 | 17.6 | 34.5 | 19.8 |
| *Ours (ExpAct)* | | | | |
| No experience completion | 50.9 | 14.1 | 42.2 | 15.3 |
| 1-round experience completion | 54.3 | 13.5 | 46.6 | 14.8 |
| 2-round experience completion | 55.2 | **12.2** | 47.4 | 15.2 |
| 3-round experience completion | **55.2** | 12.4 | **47.4** | **15.1** |

pAct is stored at the granularity of individual applications, and retrieval is restricted to experience accumulated within the target app of the current task. When no relevant experience is available, the retrieval returns an empty set and ExpAct naturally degrades to standard execution without experience augmentation.

Formally, let $\mathcal{Q}_t$ denote the embedding of the current subtask query, and $\mathcal{E}_{\text{struct}}$ and $\mathcal{E}_{\text{proc}}$ denote the embeddings of stored structural and procedural experience entries, respectively. The similarity between the query and each candidate experience entry is computed via a generalized cosine similarity function:

$$\sigma_{t,j}^{\text{struct}} = \frac{\mathcal{Q}_t \cdot \mathcal{E}_{\text{struct},j}}{\|\mathcal{Q}_t\| \|\mathcal{E}_{\text{struct},j}\|}, \quad \sigma_{t,j}^{\text{proc}} = \frac{\mathcal{Q}_t \cdot \mathcal{E}_{\text{proc},j}}{\|\mathcal{Q}_t\| \|\mathcal{E}_{\text{proc},j}\|}. \quad (5)$$

Candidate experience entries exceeding a predefined similarity threshold $\delta$ are selected, and the top-$K$ matches are retrieved for adjudication and path construction. This retrieval process can be compactly expressed as:

$$\mathcal{C}_t = \text{TopK}\Big(\{j \mid \sigma_{t,j}^{\text{struct}} > \delta\} \cup \{j \mid \sigma_{t,j}^{\text{proc}} > \delta\}\Big), \quad (6)$$

where $\mathcal{C}_t$ denotes the set of candidate experience entries provided to the LLM for final selection and structured path formation. Once the nodes and paths are determined, they are injected back into the agent's inference process to guide execution in a context-aware manner.

## 4. Experiments

### 4.1. Experimental Setting

#### 4.1.1. IMPLEMENTATION DETAILS

We implement ExpAct on two model-based mobile GUI agents, **UI-TARS-1.5-7B** (Qin et al., 2025) and **Qwen2.5-VL-7B** (Ye et al., 2025). Historical interaction trajectories generated by these agents are processed using both the LLM

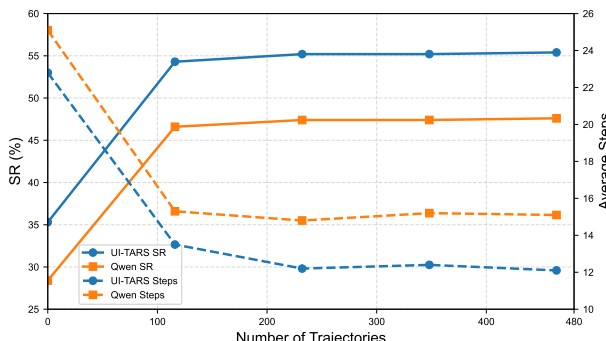

*Figure 4.* Success rate (SR, left axis) and average steps per task (right axis) on AndroidWorld, measured over increasing numbers of trajectories across multiple rounds of experience completion for two agents.

**DeepSeek-R1** (Guo et al., 2025) and the MLLM **Doubao-Seed-1.6-vision** (ByteDance, 2025) to construct, complete, and retrieve structural and procedural experience. Similarity scores for candidate selection during experience construction and retrieval are computed using the embedding model **bge-base-en-v1.5** (Xiao et al., 2023). All evaluations are performed under identical hardware, prompts, and execution settings to ensure fair comparison. We compare multiple baselines alongside ExpAct, including zero-shot execution without accumulated experience and naive experience utilization methods: *Raw Trajectory Experience*, which stores state–action sequences from zero-shot executions, and *Abstracted Text Experience*, which retains only LLM-generated step descriptions. ExpAct represents a structured, cumulative approach leveraging structural and procedural experience for subtask-aligned guidance. As shown in Fig. 4, success rate and average steps plotted against the number of trajectories indicate that the growth of the experience repository continues to yield marginal gains, with performance approaching saturation after two rounds of experience completion.

*Table 2.* Performance on AndroidWorld-LTB with UI-TARS-1.5-7B.

| Method | High-level | | Low-level | | Overall | |
|---|---|---|---|---|---|---|
| | SR ↑ | Path Eff. ↑ | SR ↑ | Path Eff. ↑ | SR ↑ | Path Eff. ↑ |
| *Zero-shot* | | | | | | |
| No Experience | 32.0 | 0.42 | 36.0 | 0.52 | 34.0 | 0.46 |
| *Ours* | | | | | | |
| ExpAct (no experience completion) | 44.0 | 0.55 | 52.0 | 0.60 | 48.0 | 0.58 |
| ExpAct (1-round experience completion) | 48.0 | 0.61 | 56.0 | 0.66 | 52.0 | 0.64 |
| ExpAct (2-round experience completion) | **48.0** | **0.62** | **60.0** | **0.67** | **54.0** | **0.65** |

*Table 3.* Task composition of the AndroidWorld-LTB benchmark. IR = Information Retrieval, Op = Operation.

| Difficulty | | Task Type | |
|---|---|---|---|
| Category | # | Category | # |
| Low-level | 25 | IR | 21 |
| High-level | 25 | Op | 29 |
| **Total Tasks** | | | **50** |

### 4.1.2. EVALUATION BENCHMARKS

**AndroidWorld.** We evaluate our method on Android-World (Rawles et al., 2024), a benchmark comprising 116 parameterized tasks across 20 real-world Android applications. AndroidWorld is utilized both to collect historical interaction trajectories, obtained by executing the base agents in a zero-shot manner, and to assess agent performance under different experience-augmented execution methods, including our proposed ExpAct. This setup allows for a systematic evaluation of how effectively ExpAct leverages accumulated experience to support task completion.

**AndroidWorld-LTB.** We introduce AndroidWorld-LTB, a specialized benchmark built upon the AndroidWorld ecosystem to evaluate ExpAct's coverage on long-tail tasks. While maintaining the standard environment dynamics and action spaces of AndroidWorld to ensure consistent evaluation, AndroidWorld-LTB is constructed as a separate benchmark comprising 50 tasks derived from the interaction space observed during execution of AndroidWorld tasks, as shown in Table 3. These tasks specifically target low-frequency widgets and sparse interaction patterns and are designed to rigorously assess the agent's knowledge retrieval and generalization capabilities under different experience-augmented methods.

### 4.2. Main Results

**Performance on AndroidWorld** We evaluate the effectiveness of different experience-augmented execution methods on AndroidWorld. Raw Trajectory Experience stores state–action sequences from zero-shot executions, while

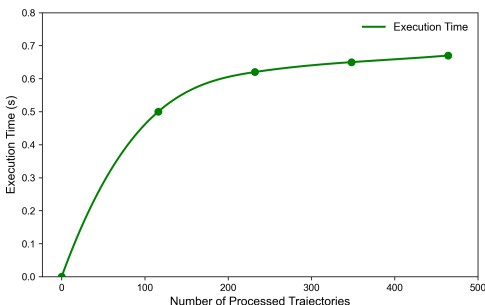

*Figure 5.* Average retrieval time per trajectory (seconds) during experience retrieval stage versus the number of processed trajectories.

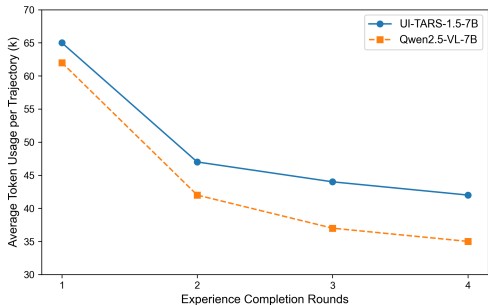

*Figure 6.* Average token usage for experience construction of two agents across different experience completion rounds.

Abstracted Text Experience retains only LLM-generated step descriptions, representing two naive episodic/textual baselines, both passively injected at task start. ExpAct consolidates historical interactions into structural and procedural experience with subtask-wise retrieval. Even without experience completion, ExpAct improves success rates and efficiency. Incorporating one or two rounds of experience completion further enriches coverage with newly successful executions, boosting guidance quality. With 3-round, UI-TARS-1.5-7B achieves 55.2% success and 12.4 steps, while Qwen2.5-VL-7B reaches 47.4% success and 15.1 steps; additional rounds yield modest gains, indicating performance stabilization. While average steps increase slightly for UI-TARS-1.5-7B after three rounds, this reflects more comprehensive experience coverage, as trajectories include additional subtask paths that improve task success rate and

*Table 4.* Ablation of experience components for UI-TARS-1.5-7B under two rounds of experience completion. SE = Structural Experience, PE = Procedural Experience, SRR = Subtask-wise Recall.

| Variant | SE | PE | SRR | SR(%) ↑ |
|---|---|---|---|---|
| Full Exp. | ✓ | ✓ | ✓ | **55.2** |
| w/o SE | ✗ | ✓ | ✓ | 48.3 |
| w/o PE | ✓ | ✗ | ✓ | 42.2 |
| w/o SRR | ✓ | ✓ | ✗ | 47.4 |

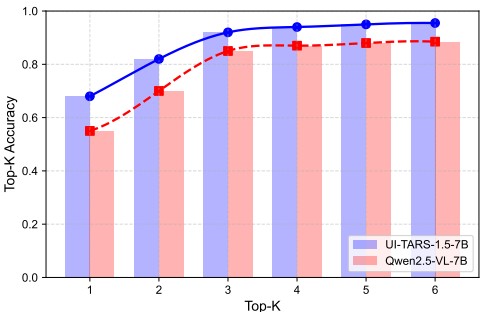

*Figure 7.* Top-K retrieval accuracy for subtask experience. Accuracy increases sharply by Top-3, then plateaus, showing diminishing gains for larger K values. Higher values indicate the agent's improved probability of retrieving correct historical subtask experience.

reduce repeated failures, and the increase remains within an acceptable range, indicating that experience exploration approaches saturation around three rounds. As Table 1 shows, ExpAct consistently delivers the highest success rates and most efficient task completion across both agents, demonstrating structured and cumulative reuse of subtask-aligned historical interactions. Results indicate that as experience grows, success rates improve and average steps decrease, highlighting the efficiency gains from experience accumulation. Fig. 7 presents the Top-$K$ retrieval accuracy for subtask-level experience retrieval. The horizontal axis corresponds to different values of $K$, while the vertical axis reports the proportion of subtask queries for which at least one ground-truth experience appears among the top-$K$ retrieved candidates. This figure is used to analyze how retrieval coverage varies as a function of the candidate set size, and to characterize the relationship between $K$ and the inclusion rate of relevant subtask experiences during retrieval.

**Performance on AndroidWorld-LTB** To evaluate robustness under sparse interaction coverage, we analyze ExpAct on AndroidWorld-LTB. As shown in Table 2, ExpAct substantially improves execution performance over zero-shot baselines even without experience completion, indicating that structured experience retrieval alone provides strong guidance in long-tail settings. Incorporating three rounds of experience completion further enhances both success rate and path efficiency, reaching 54.0% and 65.0%, respec-

tively. These results demonstrate that ExpAct consistently enhances execution performance in sparse-interaction settings, with iterative experience completion providing additional gains for long-tail tasks.

### 4.3. Overhead Analysis

We analyze the computational and token overhead of ExpAct for both the GUI agent and the MLLM generating experience. For the agent, additional cost stems from enlarged prompts due to experience injection and subtask-wise retrieval. Empirical measurements show moderate latency: after two rounds of experience completion, execution-time overhead stabilizes, with average per-trajectory retrieval adding only 0.4 ms per token (Fig. 5), indicating limited burden. For the MLLM, the main cost is processing historical trajectories to extract structural and procedural experience. This is mitigated by single-pass trajectory processing with simultaneous extraction of both experience types, and similarity-based filtering during retrieval minimizes redundant calls. As experience completion progresses, trajectory lengths shorten and redundant updates decrease, reducing per-round token consumption and generation time across successive rounds, as shown in Fig. 6).

### 4.4. Ablation Studies on ExpAct Components

We conduct ablation studies to assess the contribution of each component in **ExpAct**, including variants without structural experience, procedural experience, or subtask-wise recall. As shown in Table 4, the full ExpAct achieves a 55.2% success rate on UI-TARS-1.5-7B. Removing structural experience reduces performance to 48.3%, highlighting the importance of widget- and page-level structure. Omitting procedural experience causes a larger drop to 42.2%, indicating the central role of subtask-level procedural knowledge. Disabling subtask-wise recall lowers success to 47.4%, demonstrating the necessity of retrieving experience aligned with the current subtask. These results confirm that all components jointly contribute to ExpAct's effectiveness.

## 5. Conclusion

In this work, we presented ExpAct, a framework for experience-augmented execution in GUI agents that systematically constructs and leverages structural and procedural experience with subtask-wise recall. Experimental results show that ExpAct consistently improves task success and execution efficiency, with each component contributing to robust performance. By explicitly organizing and recalling reusable experience, ExpAct enables agents to handle sparse interactions and long-tail task variations more effectively, demonstrating the value of structured experience for enhancing generalization and reliability.

## Impact Statement

This paper presents work aimed at advancing machine learning for experience-augmented mobile GUI agents. While our framework seeks to improve efficiency and accessibility in app interactions, it carries common risks associated with autonomous agents, including potential misuse or unintended actions.

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

## A. Additional Experimental Statistics

### A.1. Error Distribution Across Completion Rounds

Fig. 8 shows a stacked bar chart illustrating the distribution of execution outcomes across different experience completion rounds. Each bar breaks down the overall outcomes into several representative categories, reflecting how different types of execution issues are distributed at each stage. The figure enables a comparative analysis of how the composition of execution outcomes evolves as experience completion progresses.

## B. Framework Details of ExpAct

We presents implementation-level details of **ExpAct** here, including prompt templates, algorithmic specifications, and auxiliary design choices that are omitted from the main paper for brevity. The goal is to enhance reproducibility and provide additional clarity on how experience is constructed, refined, and utilized in practice.

### B.0.1. STAGE I: EXPERIENCE CONSTRUCTION PROMPTS

Stage I prompts aim to transform raw interaction trajectories into reusable structural and procedural experience. The prompts focus on identifying stable interface semantics and extracting goal-driven action patterns at the subtask level.

### B.0.2. STAGE II: EXPERIENCE COMPLETION AND REFINEMENT PROMPTS

Stage II prompts support experience completion by revisiting failed or underexplored subtasks. These prompts encourage correction of misleading patterns while preserving previously successful experience as soft constraints.

### B.0.3. STAGE III: EXECUTION-AWARE RETRIEVAL PROMPTS

Stage III prompts dynamically integrate retrieved experience into the agent's inference process. The prompts are updated subtask-wise and emphasize execution relevance rather than explicit planning. We provide supplementary algorithmic details for ExpAct, complementing the high-level descriptions in the main paper. We focus on experience updates, merging rules, and retrieval criteria.

## C. Additional Case Studies

We provide additional qualitative case studies to further illustrate how experience-augmented execution alters agent behavior. These examples highlight common failure modes in zero-shot execution and demonstrate how ExpAct improves performance through experience recall.

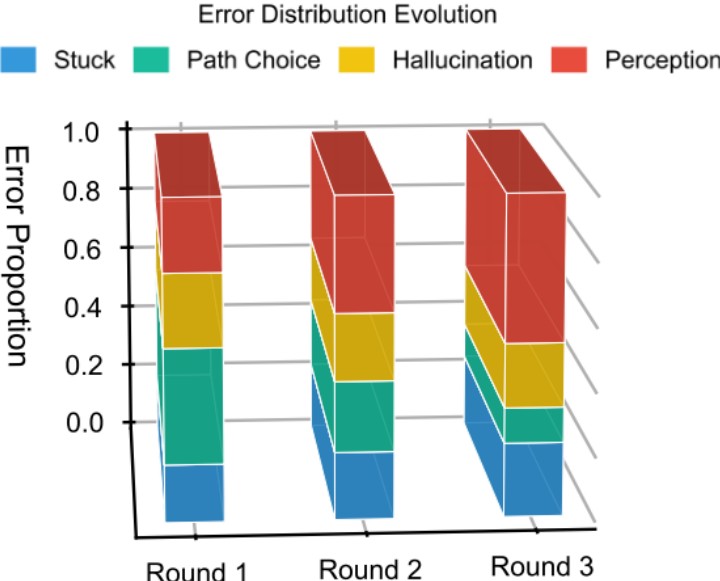

*Figure 8.* Stacked proportions of error types across experience completion rounds. Perception, path-choice, hallucination and stuck errors decrease progressively as experience is augmented, illustrating how iterative experience completion mitigates different failure modes.

```
Multi Page Comparison Prompt

"""
You are an expert in UI analysis.
Your goal is to determine which existing page (if any) matches the new page from a
    list of candidate pages.

New Page:
- Name: {new_page_name}
- Description: {new_page_description}
- Widgets: {new_page_widgets}
- Context: Triggered by action "{triggered_widget}" from page "{prev_page}"

Candidate Pages (from knowledge base):
{candidate_pages_json}

Instructions:

2. Compare the new page with each candidate page based on the following criteria:
    - Page Type Match: Popup pages can only match with popup pages; regular pages can
     only match with regular pages
    - UI Core Features: Are the component types, quantity, relative positions, and
     core text highly overlapping?
    - Interaction Context: Do they share a similar predecessor page and trigger action?
3. Determine which candidate page (if any) represents the SAME page node as the new
     page in the application graph.
4. If a match is found:
    - Return the matched page_id
    - Identify any NEW widgets in the new page that are NOT present in the matched
     candidate page.
    - Return "is_existing_page": true.
5. If no match is found:
    - Return "is_existing_page": false.
    - Return "matched_page_id": null

Output Format (JSON):
{{
    "is_existing_page": true/false,
    "matched_page_id": "page_001" or null,
    "reason": "Explanation of why a match was found or not found (mention if popup
    type mismatch)",
    "new_widgets": ["Widget1", "Widget2"]
}}
"""
```

**Widgets Deduplication Prompt**

```
"""
You are an expert in UI analysis and data cleaning.
Your task is to deduplicate and merge a list of UI widgets extracted from a mobile
    app screen.

Input:
- List of Widgets: {widgets_list}

Instructions:
1. Analyze the list of widgets.
2. Merge widgets that refer to the same element or function (e.g., "Search Icon" and
    "Search Button" -> "Search Icon").
3. Keep widgets separate ONLY if they have significant differences in function or
    important location distinctions (e.g., "Top Search Bar" vs "Bottom Search Bar").
4. Remove redundant or highly similar duplicates.
5. Standardize names to be concise and descriptive.
6. Summarize Similar Widgets: If there are multiple widgets of the same type that
    form a sequence or group, summarize them concisely.
   - Example: Instead of listing "Time Slot (08:00)", "Time Slot (09:00)", "Time Slot
    (10:00)", use "Time Slot (08:00~10:00)"
   - Example: Instead of "Day 1", "Day 2", "Day 3", use "Day (1~3)"

Output Format (JSON):
{{
    "deduplicated_widgets": ["Widget1", "Widget2", "Widget3"]
}}
"""
```

**Structural Experience Extraction Prompt**

```
"""
You are an expert in analyzing UI screens.
Your goal is to extract page information from a screenshot of a trajectory.

Input:
- Screenshot: A screenshot showing the UI state

Instructions:
1. CRITICAL - Popup/Dialog Detection: First check if a popup, dialog, modal, alert,
   or overlay is present in the screenshot
   - If a popup/dialog is present: Focus EXCLUSIVELY on the popup content. Describe
   ONLY the popup, not the underlying screen
   - If no popup/dialog: Describe the main screen as usual
   - Popups include: dialogs, modals, alerts, bottom sheets, dropdown menus, context
   menus, toast messages, permission requests, etc.
2. Identify the current page/screen name (if popup, name it as "[Popup Type]:
   [Purpose]", e.g., "Dialog: Confirm Delete", "Menu: Sort Options")
3. Provide a brief but accurate description of this page's purpose and main features
   (if popup, describe ONLY the popup)
4. Extract ALL interactive widgets visible on this page
   - If popup is present: Extract ONLY widgets within the popup, ignore the
   underlying screen
   - If no popup: Extract all widgets from the main screen
5. Identify "long_tail_widgets": These are widgets that are less prominent, hidden in
   menus, or not immediately obvious to the user. They might be specific settings,
   advanced options, or rarely used features. LIMIT: Extract between 0 and 2
   long-tail widgets maximum.
   - CRITICAL: Do NOT include common, frequently-used widgets in long_tail_widgets
   (e.g., "Close Button", "Back Button", "Save Button", "Cancel Button", "OK Button",
   "Next Button", "Previous Button", "Search Button", etc.)
   - Long-tail widgets should be truly obscure or rarely accessed features

CRITICAL RULES for widgets:
1. Extract NATIVE component types (e.g., "Button", "Text Field", "Icon", "Checkbox",
   "Menu Item")
2. Do NOT extract dynamic content or specific values (e.g., use "Note Item" not
   "Booking Note", "Task Item" not "Buy Groceries Task")
3. Focus on UI elements that represent interaction points, not content
4. Extract ONLY static, app-native widgets that are likely to persist across
   different sessions
5. If a popup/dialog is present: Extract ONLY widgets from the popup, completely
   ignore the underlying screen
6. Summarize Similar Widgets: If there are multiple widgets of the same type that
   form a sequence or group, summarize them concisely.
   - Example: Instead of listing "Time Slot (08:00)", "Time Slot (09:00)", "Time Slot
   (10:00)", use "Time Slot (08:00~10:00)"
   - Example: Instead of "Day 1", "Day 2", "Day 3", use "Day (1~3)"

Output Format (JSON):
{{
    "page_context": "Current page/screen name (if popup, use format: 'Dialog:
    Purpose' or 'Menu: Purpose')",
    "page_description": "Brief but accurate description of this page's purpose and
    main features (if popup, describe ONLY the popup)",
    "widgets": ["Widget1", "Widget2", "Widget3"],
    "long_tail_widgets": ["LongTailWidget1"],
    "is_popup": true/false
}}
"""
```

```
Procedural Experience Extraction Prompt

"""
You are an expert in analyzing UI automation trajectories.
Your goal is to extract task-related subtask path knowledge as reusable templates.

Input:
- App Name: {app_name}
- Task Description: {task_description}
- Execution Data: Page descriptions, step descriptions, and interacted widgets

Instructions:

1. Subtask Extraction (Task-Related Only)
- Extract only subtasks directly required to accomplish the given task
- Use a single subtask for simple tasks; split into multiple subtasks only when the
    task has clear phases
- Remove redundant actions; deduplicate equivalent paths

2. Template-Based Path Representation
- Represent subtask paths as templates with placeholders: {{placeholder_name}}
- Replace all concrete values (names, queries, numbers)
- Keep UI actions and control names concrete
  (e.g., "Tap search button", "Enter {{query}} in search field")

3. Step Specification
- Each step must describe ONLY:
  (a) the action, and (b) the interacted control
- Avoid navigation-level or abstract descriptions

4. Context Specification
- Provide a concise context to locate the control:
  widget location, surrounding elements, and page function
- Maximum three sentences
- Do NOT include a separate widget field

5. Misleading Paths
- Include only paths easily confused with correct subtask paths
- Each misleading step contains description only (no context)
- Deduplicate misleading paths; leave empty if none exist

6. Denoising Rules
- Focus on native UI components (Button, Text Field, Icon, Menu Item)
- Ignore dynamic or instance-specific content
- Remove unnecessary back-and-forth navigation

Output Format (JSON):
{
  "subtask_paths": [
    {
      "subtask_name": "...",
      "success_paths": [
        { "steps": [
            { "description": "...", "context": "..." }
        ]}
      ],
      "misleading_paths": [
        { "steps": [
            { "description": "..." }
        ], "error_reason": "..." }
      ]
    }
  ]
}

16

"""
```

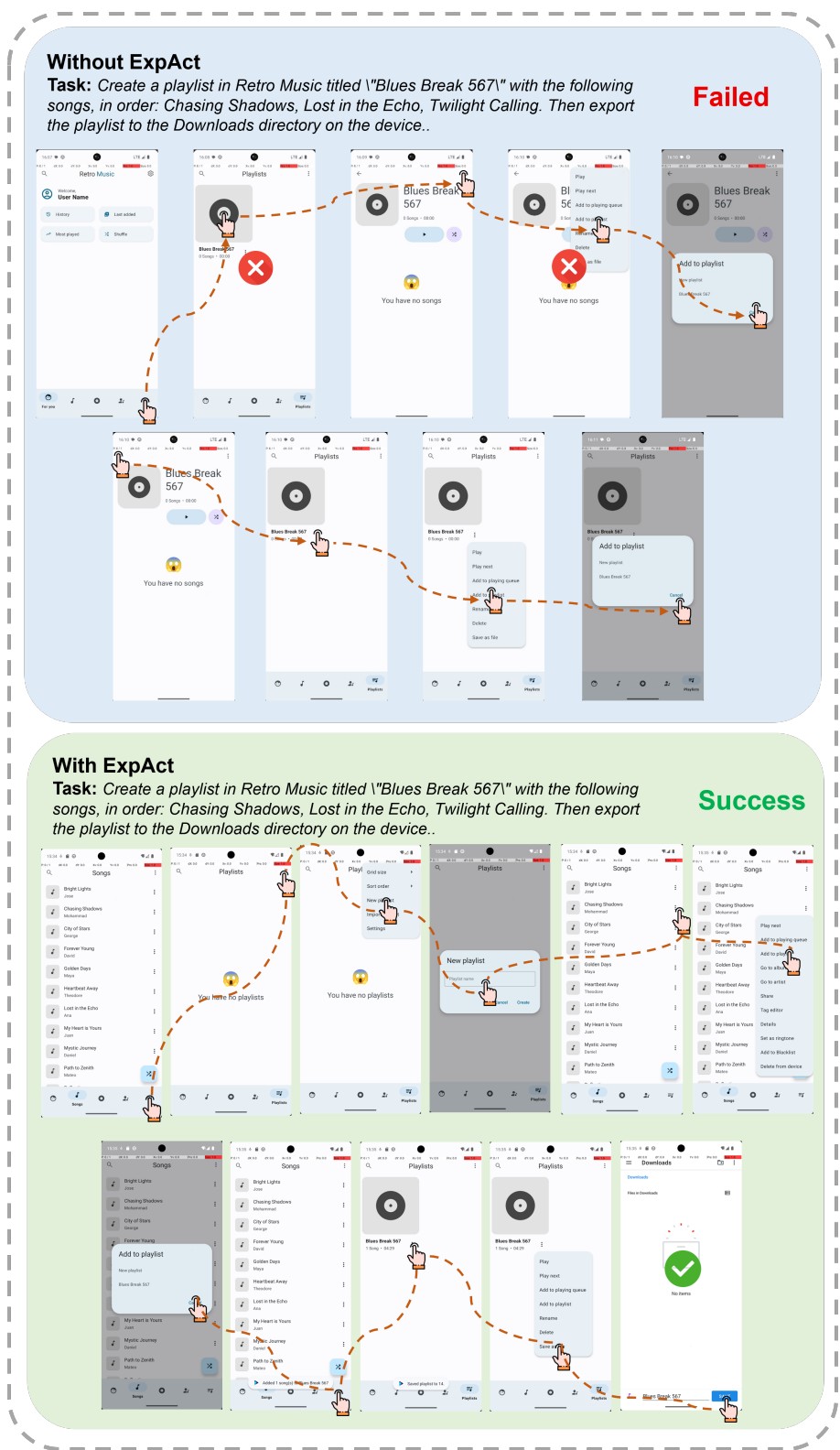

*Figure 9.* Execution trajectory comparison of the "create and export playlist" task on Retro Music with/without ExpAct. The upper panel (Without ExpAct) shows the agent failing to complete the task due to invalid repeated operations and misnavigation, while the lower panel (With ExpAct) demonstrates successful task execution via guided experience retrieval and completion, resulting in correct playlist creation and export to the Downloads directory.

