# OpenReview forum: "ExpAct: Building and Evolving Structural and Procedural Experience for Mobile GUI Agents"
_ICML.cc/2026/Conference — Submitted to ICML 2026_

### Official Review · Reviewer_Nza5 · 2026-03-08

**Soundness:** 3
**Presentation:** 3
**Significance:** 3
**Originality:** 3
**Overall Recommendation:** 5
**Confidence:** 4

**Summary:**

This paper introduces ExpAct, a training-free, experience-centric framework designed to help Multimodal Large Language Model (MLLM) GUI agents accumulate, refine, and reuse interaction experience over time. The authors address the limitation that current GUI agents typically execute tasks in an episodic, zero-shot manner, causing them to frequently re-explore known interfaces and repeat identical errors.

To solve this, ExpAct leverages the agent's own historical interaction trajectories across three distinct stages:

Stage 1: Experience Construction. Raw execution traces are abstracted into two complementary representations: a "Structural Experience" (a directed graph mapping UI pages, semantics, and widget transitions) and a "Procedural Experience" (subtask-level action sequences detailing successful paths and misleading paths).

Stage 2: Experience Completion and Refinement. The framework actively re-executes failed or underexplored tasks to correct misleading paths, fill knowledge gaps, and progressively enrich the structural graph.

Stage 3: Execution-Aware Experience Retrieval. During runtime, tasks are decomposed into subtasks, and the framework uses vector similarity to retrieve relevant structural and procedural knowledge to dynamically guide the agent's prompt without explicit retraining.

The authors evaluate ExpAct on the AndroidWorld benchmark using UI-TARS-1.5-7B and Qwen2.5-VL-7B, demonstrating notable improvements in task success rates and step efficiency compared to zero-shot execution and naive textual memory baselines. Additionally, the paper introduces a new 50-task benchmark, AndroidWorld-LTB, to specifically evaluate agent performance on low-frequency widgets and sparse interaction patterns.

**Compliance With Llm Reviewing Policy:**

Affirmed.

**Final Justification:**

The authors provided a strong rebuttal that directly addressed my initial concerns.

Most notably, the inclusion of the Mobile-Agent-E baseline empirically validates that ExpAct outperforms purely textual hierarchical memory systems on long-horizon tasks. Furthermore, the clarification on the statistical thresholds used for AndroidWorld-LTB (bottom 10% occurrence, top 10% path length) and the additional experiments on AndroidLab successfully resolve my concerns regarding benchmark bias and cross-environment transferability.

I also appreciate the authors engaging with the interaction mining literature (ODIM and ERICA) as a compatible extension for future work, and fixing the notational collisions in Section 3.

Because the authors have successfully closed the gaps in their evaluation and improved the paper's formal rigor, I am raising my score to an Accept. I expect the authors to incorporate the new baselines, clarifications, and notation fixes into the camera-ready version.

**Key Questions For Authors:**

1. Baseline Comparisons with Existing Memory Frameworks:
Why was Mobile-Agent-E excluded from the empirical baselines in Section 4? Given that it also utilizes a dynamic, long-term memory retrieval system (via "Shortcuts" and "Tips"), it serves as a highly relevant state-of-the-art comparison.

2. Integrating Human Interaction Traces:
Currently, ExpAct relies entirely on zero-shot agent self-exploration to bootstrap its experience (Stage 1). How would the framework perform if seeded with high-fidelity, human-generated interaction traces collected from the wild, such as those mined by frameworks like ODIM [1] or ERICA [2]?

3. Robustness and Bias in AndroidWorld-LTB:
With only 50 tasks in the AndroidWorld-LTB benchmark, how did the authors ensure these tasks represent a generalized "long-tail" rather than being inadvertently cherry-picked to favor scenarios where structural widget graphs naturally excel?

4. Resolving Notational Collisions:
Will the authors commit to resolving the variable overloading in Section 3 (specifically, the dual use of $\mathcal{E}$ for both the experience repository and graph edges, and the dual use of $\mathcal{S}$ for both the state space and subtask set)?

**Limitations:**

Yes

**Strengths And Weaknesses:**

Strengths

Soundness:

- Comprehensive Ablations: The ablation studies effectively prove the necessity of the dual-layered abstraction. The significant performance drop when removing the procedural experience strongly validates the decision to separate spatial interface mapping from operational task trajectories.

- Operational Efficiency: The overhead analysis (latency and token costs) is a strong addition. Showing that token usage actually decreases across experience completion rounds—because the agent learns to avoid misleading paths—convincingly demonstrates that the system does not computationally degrade as the memory bank grows.

Presentation:

- Intuitive Architecture: The conceptual separation of "Structural Experience" (the interface map) and "Procedural Experience" (the task-oriented paths) is highly logical and well-explained, making the core mechanism easy to grasp.

Significance:

- Practical Error Correction: The paper addresses a highly relevant limitation in current GUI agents—episodic amnesia. The "Experience Completion" loop (Stage 2) provides a practical, training-free retrieval mechanism that actively corrects specific failure modes like perception errors and hallucinated paths.

Originality:

- Graph-Based RAG for GUIs: Converting linear execution traces into a structured, queryable widget graph, combined with subtask-wise dynamic retrieval, is a creative and highly practical application of retrieval-augmented generation to graphical interface navigation.

------------------

Weaknesses

Soundness:

- Missing State-of-the-Art Baselines: A critical flaw in the experimental setup is the omission of existing memory-augmented agent baselines. The authors cite Mobile-Agent-E in their related work—a framework that also builds long-term operational memory via reusable "Shortcuts" and "Tips"—but fail to benchmark against it. Comparing ExpAct only against zero-shot and naive text-dump baselines leaves it unclear whether the proposed widget graph approach empirically outperforms existing hierarchical memory systems using the same base models (UI-TARS or Qwen).

- Scope of the Custom Benchmark: While the introduction of the AndroidWorld-LTB benchmark is a well-motivated attempt to test agent robustness under sparse interaction patterns, a sample size of only 50 tasks is arguably too small to draw statistically definitive conclusions about the framework's generalized long-tail performance.

Originality & Significance:

- Omission of Interaction Mining Literature: The paper frames its "experience-centric perspective" as a primary contribution. However, it relies entirely on zero-shot LLM self-exploration to bootstrap its trajectories. The authors completely overlook the extensive HCI and systems literature on on-device interaction mining frameworks that capture high-fidelity, human-generated traces in the wild (such as ODIM [1] and ERICA [2]). Acknowledging these works and discussing how abstract "experiences" compare to raw human "traces" would significantly strengthen the paper's theoretical grounding and suggest more robust ways to seed the Stage 1 experience construction.

Presentation:

- Mathematical Notation Collisions: The formalization in Section 3 suffers from severe variable overloading. For instance, in Equation 3, the symbol $\mathcal{E}$ is used to represent the edges of the widget graph, colliding directly with its prior definition as the experience repository ($\mathcal{E}_t$) in Section 3.1. Similarly, $\mathcal{S}$ is used to denote the set of subtasks ($g \in \mathcal{S}$), despite being defined earlier as the environment state space. These collisions break mathematical continuity and should be resolved.

- Overstated Contribution Framing: Elevating an "experience-centric perspective" to a top-level contribution overstates the conceptual novelty. The true contribution lies in the specific architectural mechanisms (the widget graph and subtask retrieval pipeline) , not the general viewpoint of reusing past data, which is standard across reinforcement and continuous learning paradigms.


[1] https://doi.org/10.1145/3743726
[2] https://doi.org/10.1145/2984511.2984581

---

> ### Author Rebuttal · Authors · 2026-03-30
>
> Thank you for your thorough review and constructive feedback on our paper.
>
> >**W1&Q1: Comparison with Existing Memory-Based Agents**
>
> We thank the reviewer for this important suggestion. We agree that including Mobile-Agent-E is a valuable comparison, and we have now incorporated it into our evaluation under the same base agents and experimental setup.
>
> As shown in Table 1, ExpAct consistently achieves higher success rates across multiple rounds of evolution, indicating that ExpAct provides more effective guidance for long-horizon tasks. These results suggest that textual hierarchical memory alone is insufficient to fully capture reusable GUI interaction structure, whereas ExpAct’s **structured experience representation and execution-aware, subtask-level retrieval** provide more effective guidance for long-horizon tasks.
>
> Table 1: Comparison with Mobile-Agent-E on AndroidWorld
> |Method|UI-TARS-1.5-7B|Qwen2.5-VL-7B|
> |-|-|-|
> ||SR(%)↑|SR(%)↑|
> |Ours (ExpAct) Evo-1|54.3|46.6|
> |Mobile-Agent-E Evo-1|42.2|31.9|
> |Ours (ExpAct) Evo-2|55.2|47.4|
> |Mobile-Agent-E Evo-2|42.2|31.9|
> >**W2&Q3: Benchmark scale and long-tail representativeness**
>
> AndroidWorld-LTB is motivated by the **long-tail nature** of mobile interactions, where low-frequency cases are critical yet often responsible for failures.  It serves as a controlled stress test targeting **sparse interaction regimes**, rather than being inadvertently cherry-picked to favor structural widget graph scenarios. The 50 tasks are selected based on trajectory statistics, focusing on UI elements whose occurrence falls in the bottom 10% and subtask paths whose length falls in the top 10% of the trajectory distribution, ensuring that they reflect genuinely underexplored regions rather than favorable cases.
>
> To further validate generality beyond the AndroidWorld ecosystem, we additionally provide results on the AndroidLab benchmark for both agents below, demonstrating that ExpAct effectively transfers to other environments.
>
> Table 2. Performance on AndroidLab benchmark
> |Method|UI-TARS-1.5-7B||Qwen2.5-VL-7B||
> |-|-|-|-|-|
> ||SR(%)↑|Avg.Steps↓|SR(%)↑|Avg.Steps↓|
> |**Zero-shot**|||||
> |No Experience|23.2|24.1|10.1|28.3|
> |**Ours (ExpAct)**|||||
> |No experience completion|35.5|17.7|24.6|21.8|
> |1-round experience completion|38.4|15.9|27.5|19.4|
> |2-round experience completion|39.1|16.2|27.5|19.2|
> |3-round experience completion|**39.9**|**15.3**|**28.3**|**18.6**|
>
> >**W3&Q2: Human Interaction Trace Integration**
>
> Thank you for highlighting interaction mining work. While human-generated traces are valuable, our focus is on a **self-accumulating and self-refining paradigm**, where the agent autonomously builds and updates its experience without external data pipelines. In contrast, human-trace–based approaches often require costly collection and maintenance and may become brittle as interfaces evolve. ExpAct instead continuously adapts through exploration and iterative completion. Incorporating human traces is not the core objective but a compatible extension, we view this as a promising direction for future work.
>
> >**W4&Q4: Notation consistency**
>
> We thank the reviewer for pointing out this issue. We have revised the notation to remove symbol overloading and make the definitions more precise. Specifically, we replace the experience notation $\mathcal{E}$ with $E$, and the subtask set $S$ with $\mathcal{U}$.
> The updated formulation is given as follows:
> $$
> E_{\text{}}(\mathcal{T})
> = \Big( \mathcal{G}(\mathcal{V}, \mathcal{E}, \Phi_s, \Phi_w),
> \bigcup_{g \in \mathcal{U}} \{ (g, \tau_g, y_g, r_g) \} \Big)
> $$
>
> >**W5: Contribution framing**
>
> We agree that reusing past experience is a general paradigm across reinforcement and continual learning, and the **widget graph and subtask retrieval pipeline** are key contributions of our work. Our intention in describing an “experience-centric” perspective is to highlight **a unified framework where experience is represented, improved, and used during execution**, rather than the general idea of experience reuse. Specifically, ExpAct maintains a dual-form experience store with widget-level structural graphs and procedural patterns capturing success and failure, providing a richer representation than static text (RAG) or abstract relations (KG); it iteratively accumulates and refines experience by revisiting and correcting failure cases, enabling dynamic improvement beyond fixed experience replay; and it applies experience through subtask-wise retrieval, allowing more fine-grained retrieval and application. Overall, our contribution lies in a unified experience-centric framework for representing, refining, and applying experience.

---

> > ### Author Rebuttal · Reviewer_Nza5 · 2026-04-03
> >
> > Thank you for the comprehensive rebuttal. My concerns have been fully addressed.
> >
> > * **Baselines:** Running the Mobile-Agent-E comparison under the same setup resolves my primary concern regarding efficacy. The results clearly demonstrate the advantage of structured widget graphs over purely textual memory.
> > * **Benchmark Robustness:** Providing the explicit statistical thresholds for AndroidWorld-LTB (bottom 10% occurrence, top 10% path length) effectively removes concerns about structural bias. The added AndroidLab evaluation is also a strong proof of cross-environment transfer.
> > * **Human Traces:** I agree that integrating human traces (e.g., ODIM/ERICA) is best positioned as a compatible extension for future work, and I appreciate the acknowledgment of this literature.
> > * **Notation:** Thank you for resolving the variable overloading in the formalization ($\mathcal{E} \rightarrow E$ and $\mathcal{S} \rightarrow \mathcal{U}$).
> >
> > I will raise my score. Please ensure the Mobile-Agent-E baseline, AndroidLab results, updated math notation, and the brief discussion of the interaction mining literature are included in the camera-ready version.

---

> > > ### Author Response · Authors · 2026-04-05
> > >
> > > Dear Reviewer Nza5,
> > >
> > > Thank you for your time and effort in reviewing our work. We are glad that our responses and additional experiments have addressed your concerns, and appreciate your insightful feedback.
> > >
> > > In the revision, we will include the Mobile-Agent-E comparison (Table 1), additional AndroidLab results (Table 2), update the mathematical notation, and include a brief discussion of interaction mining literature.
> > >
> > > We also appreciate your positive assessment and your willingness to raise the score. All these updates will be included in the final version.
> > >
> > > Thank you again for your thoughtful evaluation.
> > >
> > > Best regards,
> > >
> > > The Authors

---

### Official Review · Reviewer_Qotb · 2026-03-12

**Soundness:** 3
**Presentation:** 3
**Significance:** 3
**Originality:** 3
**Overall Recommendation:** 5
**Confidence:** 3

**Summary:**

The paper proposes ExpAct, an experience-centric framework for mobile GUI agents that accumulates and organizes historical interaction trajectories into structural interface knowledge and procedural subtask patterns. The framework retrieves relevant experience during execution to guide the agent and avoid repeated mistakes. Experiments on AndroidWorld show that ExpAct improves task success rate and reduces the number of steps required for task completion compared to zero-shot and naive experience baselines.

**Compliance With Llm Reviewing Policy:**

Affirmed.

**Final Justification:**

Strong paper and author's rebuttal even addressed some of my minor concerns.

**Key Questions For Authors:**

See the weakness above.

**Limitations:**

no discussion about limitations

**Strengths And Weaknesses:**

Strength

1. The paper is well written and clearly motivated, with a clear discussion of the limitations of purely episodic GUI agents and the need for accumulating reusable interaction experience.
2. The proposed framework introduces an experience-centric perspective for GUI agents by explicitly modeling both structural interface knowledge and procedural interaction patterns, which is a meaningful and intuitive design.
3. The experiments demonstrate consistent improvements in success rate and task efficiency, and the paper includes additional analyses such as ablations and overhead measurements that help understand the contribution of different components.

Weakness

1. While the method reduces the average number of steps required to complete tasks, the paper does not report end-to-end latency comparisons between the baseline agent and the experience-augmented system. Since ExpAct introduces additional retrieval and prompt processing, it remains unclear how these factors affect overall execution time.
2. The evaluation focuses on experience reuse across different tasks within the same application, while the ability of the experience representation to transfer across applications or more diverse environments is not explored.

---

> ### Author Rebuttal · Authors · 2026-03-31
>
> Thank you for your insightful review and strong support for our work. We greatly appreciate your recognition of our motivation, experience-centric design, and empirical analyses. Below are our detailed responses to the points you raised.
>
> >**W1: End-to-end latency**
>
> We agree that end-to-end latency is an important complement to step efficiency. We already analyze retrieval time across increasing numbers of processed trajectories in Fig. 5 of our paper, showing that the overhead remains moderate and stabilizes as experience becomes more complete, which averages only 0.5~0.7 seconds per task. This accounts for a small fraction of total runtime, while the reduction in execution steps leads to a clear overall efficiency gain. Moreover, retrieval latency can be further reduced with more efficient retrieval models and LLM acceleration. Overall, the method introduces limited overhead while effectively reducing redundant exploration and repeated failures.
>
> To make this more explicit, we report end-to-end latency on AndroidWorld with UI-TARS-1.5-7B, as shown in Table 1. The results include total execution time per task together with average steps, showing that as experience completion progresses, the overall runtime is consistently reduced and remains stable.
>
> Table 1. End-to-end latency analysis on AndroidWorld.
>
>
> |Method|SR (%) ↑|Avg. Steps ↓|Avg. Runtime / Task (s) ↓|
> |-|-|-|-|
> |Zero-shot|35.3|22.8|58.4|
> |ExpAct (no completion)|50.9|14.1|36.5|
> |ExpAct (1-round completion)|54.3|13.5|33.7|
> |ExpAct (2-round completion)|55.2|12.2|30.6|
>
> >**W2: Transfer across applications and environments**
>
>
>
> Thank you for raising this point. We agree that broader transfer is an important direction, while our current evaluation focuses on experience reuse within the same application, which **aligns with our problem setting and realistic usage scenarios** where agents repeatedly interact with the same apps and progressively build reusable experience.
>
> Transferring experience across applications is an appealing direction, but also involves additional challenges such as differences in UI organization, interaction patterns, and task structure. In this work, we prioritize understanding experience reuse within a single-application setting, where these factors remain stable and the effect of experience can be more clearly evaluated. We therefore treat cross-application transfer as an important but separate problem that is not the primary focus of this work due to space constraints, and consider it a promising direction for future work.
>
> For transfer across diverse environments, we provide additional results on the AndroidLab benchmark for both agents below. As shown in Table 2, ExpAct effectively generalizes beyond the AndroidWorld ecosystem, indicating that our framework remains effective across different environments.
>
> Table 2. Performance on AndroidLab benchmark
>
> |Method|UI-TARS-1.5-7B||Qwen2.5-VL-7B||
> |-|-|-|-|-|
> ||SR (%) ↑|Avg. Steps ↓|SR (%) ↑|Avg. Steps ↓|
> |**Zero-shot**|||||
> |No Experience|23.2|24.1|10.1|28.3|
> |**Ours (ExpAct)**|||||
> |No experience completion|35.5|17.7|24.6|21.8|
> |1-round experience completion|38.4|15.9|27.5|19.4|
> |2-round experience completion|39.1|16.2|27.5|19.2|
> |3-round experience completion|**39.9**|**15.3**|**28.3**|**18.6**|

---

> > ### Author Rebuttal · Reviewer_Qotb · 2026-04-08
> >
> > Thanks for authors's rebuttal. I will keep my score, which means that I still think this is a strong paper and we should accept it.

---

> > > ### Author Response · Authors · 2026-04-08
> > >
> > > Dear Reviewer Qotb,
> > >
> > > Thank you for your time and effort in reviewing our work. We are glad that our responses and additional experiments have addressed your concerns, and appreciate your insightful feedback.
> > >
> > > In the revision, we will include a detailed analysis of end-to-end latency (Table 1), provide additional AndroidLab results (Table 2), and further clarification on the framework design. All these updates will be included in the final version.
> > >
> > > Thank you again for your thoughtful evaluation of our work.
> > >
> > > Best regards,
> > >
> > > The Authors

---

### Official Review · Reviewer_rzC7 · 2026-03-13

**Soundness:** 4
**Presentation:** 3
**Significance:** 4
**Originality:** 4
**Overall Recommendation:** 4
**Confidence:** 4

**Summary:**

A general area explored by this article is long-term, reusable experience for GUI agents operating mobile apps. The work proceeds to address a notable topic: how to move beyond episodic, zero-shot operation toward agents that accumulate, structure, and reuse their own interaction history.

The paper proposes ExpAct, an “experience layer” that sits on top of existing mobile GUI agents and:

Builds a structural representation of app interfaces as a widget-level graph.
Extracts subtask-level procedural patterns (both successful and failed) from past trajectories.
Iteratively refines this experience by targeted re-execution of failed or underexplored cases.
Retrieves subtask-aligned experience at execution time to guide the base agent without changing its policy.
It further introduces AndroidWorld-LTB, a long-tail benchmark derived from AndroidWorld to stress-test performance on low-frequency widgets and sparse interaction patterns. Experiments with two existing agents show sizable gains in success rate and efficiency, especially on long-tail scenarios, and ablations indicate that both structural and procedural components contribute.

**Compliance With Llm Reviewing Policy:**

Affirmed.

**Key Questions For Authors:**

Clarify the novelty and generality of the framework

More explicitly contrast ExpAct with prior retrieval-augmented and knowledge-graph approaches: what can ExpAct do that they fundamentally cannot, beyond using widget graphs and subtask templates?
Discuss whether the same framework could be applied to non-GUI sequential decision problems, or whether it is inherently domain-specific.


Robustness to noisy experience construction

How often do the LLM-based abstractions (page matching, widget deduplication, subtask segmentation) misfire, and how does that propagate into retrieval-time errors?
A controlled experiment where synthetic noise is injected into the experience store would strengthen the robustness claims.


Scalability and pruning mechanisms

Provide a more explicit strategy for pruning or compressing experience as it grows: e.g., forgetting rarely-used, low-utility paths; clustering similar subtask patterns; or compressing the graph.
Reporting memory usage and retrieval latency for a larger-scale setting (more apps, more trajectories) would be very informative.


Benchmark design and cross-environment evaluation

Give a more formal description of how “low-frequency widgets” and “sparse interaction patterns” are defined when constructing AndroidWorld-LTB; this will matter for reproducibility and fairness.
If possible, evaluate ExpAct on at least one additional benchmark (e.g., another Android or cross-platform environment) to test generalization beyond the AndroidWorld ecosystem.


Understanding retrieval decisions and conflicts

The system may confront conflicting experiences (e.g., multiple paths with different outcomes for similar subtasks). How are these adjudicated beyond simple semantic similarity and LLM adjudication?
Adding qualitative analyses of such conflict cases, and perhaps tracking how often negative experience successfully overrides misleading positive examples, would add depth.


Ablation on experience completion

The iterative completion procedure is empirically helpful, but its mechanics remain high-level.
A more fine-grained analysis of what is gained in each completion round (e.g., diversity of new structural edges, subtask variants discovered) could better justify this part of the pipeline.

**Limitations:**

The paper is solidly executed, empirically convincing, and tackles a practically important problem—structured long-term experience for GUI agents—with a reasonably novel systems design and a useful benchmark contribution. The main reservations are that the ideas are somewhat close to existing RAG/experience-replay paradigms and that scalability and robustness are not fully resolved. Nonetheless, for ICLR, the combination of clear problem framing, methodical system design, and strong empirical gains, especially on long-tail tasks, justifies acceptance. The only pitfall: Limited analysis of failure modes — While error-type distributions are reported, the discussion remains mostly descriptive. There is little probing of where the method still fails (e.g., distribution shift in interface design, incorrect subtask inference, conflicting experiences) and how those failure modes might be mitigated.

**Strengths And Weaknesses:**

Strengths

Clear experience-centric framing

The paper articulates a coherent shift from episodic execution to structured experience accumulation and reuse, which is under-explored in GUI agents.
The decomposition into structural and procedural experience is conceptually clean and maps well to the GUI domain.

System design that is modular and agent-agnostic

ExpAct is explicitly designed as an add-on “experience layer” that does not require retraining or modifying the underlying GUI agent, making it broadly applicable.
The subtask-wise retrieval mechanism is a sensible way to align past experience with current execution progress.

Handling both positive and negative experience

Treating failed trajectories as negative constraints, rather than discarding them, is a strong design choice and aligns with how real systems avoid repeating past mistakes.

Long-tail benchmark contribution

AndroidWorld-LTB addresses an important evaluation gap: how agents behave on low-frequency widgets and sparse interaction paths, instead of only on frequent, “head” interactions.
The benchmark is systematically derived from an existing suite, providing some reproducibility and relevance.

Empirical evidence and ablations

The method is demonstrated across multiple base agents, showing that the gains are not specific to a single model.
Ablation over structural vs procedural experience and retrieval variants provides reasonable support for the claimed design choices.
Overhead analysis suggests the approach is at least plausibly practical.



Weaknesses

Limited conceptual novelty relative to RAG/experience replay

The core idea—building a structured memory from trajectories and doing retrieval at execution time—is closely related to retrieval-augmented generation and experience replay.
The paper does not fully sharpen what is fundamentally new beyond adapting these ideas to the GUI context with some domain-specific engineering.

Heavy reliance on external LLMs and heuristics

Many key operations (page abstraction, subtask decomposition, failure labeling, experience completion prompts, etc.) depend on multiple large models and hand-crafted prompt pipelines.
Robustness to errors in these annotations is not deeply analyzed, and the approach may be brittle if upstream LLM quality drops or prompts change.

Unclear scalability and maintenance story

As experience grows across many apps and tasks, it is not fully clear how the repository is kept compact, de-duplicated, and efficient.
The paper shows retrieval overhead trends, but does not convincingly address long-term scaling (e.g., hundreds of apps, persistent deployment).

Benchmark construction and generality

AndroidWorld-LTB is derived from the same environment used to build experience, which risks evaluation being tailored to the proposed method’s strengths.
It is not fully clear how representative these “long-tail” tasks are of real-world mobile usage, nor how well the approach would transfer to other environments or platforms.

---

> ### Author Rebuttal · Authors · 2026-03-30
>
> Thank you for your thorough review and positive feedback on our paper.
>
> >**W1&Q1: The novelty and generality of our framework**
>
> **Compared to RAG, experience replay, and knowledge-graph approaches, our contribution lies in how experience is represented, improved, and used during execution.** Specifically, ExpAct maintains a dual-form experience store with widget-level structural graphs and procedural patterns capturing success and failure, providing a richer representation than static text (RAG) or abstract relations (KG); it iteratively accumulates and refines experience by revisiting and correcting failure cases, enabling dynamic improvement beyond fixed experience replay; and it applies experience through subtask-wise retrieval, allowing more fine-grained retrieval and application.
>
> Regarding **generality**, our design is GUI-specific, but we believe the paradigm of **experience representation and iterative refinement** can extend to other sequential decision problems.
>
> >**W2&Q2: Robustness to noisy experience**
>
> We agree that, given the inherent uncertainty in LLM-based abstraction, some degree of noisy experience may arise. However, ExpAct shows consistent improvements across completion rounds with task success rate increasing, indicating stable performance even with potentially noisy experience. Each round expands coverage and corrects failure-prone cases as shown in Table 3, suggesting that errors do not accumulate but are mitigated through iterative refinement over time.
>
> >**W3&Q3: Scalability and maintenance mechanisms**
>
> We appreciate this concern. **Our design supports scalability through similarity-based consolidation and per-application storage**, as described in Section 3.2. Concretely, we use a similarity-based self-check with LLM adjudication to cluster similar subtask patterns and compress the graphs, and store experience per application to keep retrieval efficient.
>
> We report **memory usage** and **retrieval latency** for a larger-scale setting, as shown in Table 1. The results show that both memory and latency remain well-bounded and can be further improved through several strategies. For memory, the retrieval scope can be reduced through user-level or scenario-level partitioning to improve efficiency. For retrieval, latency can be further optimized with faster retrieval models and LLM acceleration.
>
> Table 1: Analysis on memory usage and retrieval latency
> |Trajectories|Memory Usage(MB)|Retrieval Time per Traj(s)|
> |-|-|-|
> |116|1.3|0.50|
> |464|3.4|0.67|
> |878|5.3|0.74|
>
> >**W4&Q4: Benchmark design and cross-environment evaluation**
>
> Thank you for this suggestion. We clarify the definitions as follows:
> * Low-frequency widgets: UI elements whose occurrence frequency falls in the bottom 10% across trajectories.
> * Sparse interaction patterns: subtask paths whose step length falls in the top 10% of the trajectory length distribution.
>
> AndroidWorld-LTB is motivated by the long-tail nature of mobile interactions, where low-frequency cases are critical yet often responsible for failures. We provide additional results on the AndroidLab benchmark for both agents below, demonstrating that ExpAct effectively transfers to other environments.
>
> Table 2: Performance on AndroidLab benchmark
> |Method|UI-TARS-1.5-7B||Qwen2.5-VL-7B||
> |-|-|-|-|-|
> ||SR (%) ↑|Avg. Steps ↓|SR (%) ↑|Avg. Steps ↓|
> |**Zero-shot**|||||
> |No Experience|23.2|24.1|10.1|28.3|
> |**Ours (ExpAct)**|||||
> |No completion|35.5|17.7|24.6|21.8|
> |1-round completion|38.4|15.9|27.5|19.4|
> |2-round completion|39.1|16.2|27.5|19.2|
> |3-round completion|**39.9**|**15.3**|**28.3**|**18.6**|
>
> >**Q5: Retrieval conflicts and decision**
>
> ExpAct resolves conflicts through **context alignment and fallback-based exploration of alternative paths**. For a subtask such as “take a video,” two similar paths may be retrieved: one that switches to video mode and one that does not. ExpAct injects subtask-aware context to guide selection toward the path that includes video mode, and even if an incorrect path is selected initially, execution feedback will trigger fallback to alternative correct candidates.
> We analyze how negative experience improves decision quality by measuring the fraction of failed subtasks corrected across completion rounds, as shown in Table 3, and observe that it leads to improved success rates over time.
>
> >**Q6: Fine-grained analysis of experience completion**
>
> Thank you for this suggestion. We analyze the experience repository on AndroidWorld with UI-TARS-1.5-7B, tracking changes in structural and procedural experience across rounds. Results in Table 3 show that early rounds introduce major coverage gains and corrections, while later rounds provide smaller, consistent refinements and stabilize performance.
>
> Table 3: Experience evolution across completion rounds.
> |Round|Pages|Widgets|Pos Patterns|Neg Patterns|SR|
> |-|-|-|-|-|-|
> |Init|112|862|158|190|50.9|
> |R1|146|1095|220|136|54.3|
> |R2|157|1160|243|119|55.2|
> |R3|160|1183|255|109|55.2|

---

> > ### Author Rebuttal · Reviewer_rzC7 · 2026-04-03
> >
> > I have read the rebuttal and maintain the same positive rating of this work.

---

> > > ### Author Response · Authors · 2026-04-05
> > >
> > > Dear Reviewer rzC7,
> > >
> > > Thank you for your time and effort in reviewing our work. We are glad that our responses and additional experiments have addressed your concerns, and appreciate your positive feedback.
> > >
> > > In the revision, we will expand the analysis of memory usage and retrieval latency (Table 1), include additional AndroidLab results (Table 2), more fine-grained analysis of experience completion (Table 3), and additional clarification on the framework design. All these updates will be included in the final version.
> > >
> > > Thank you again for your commitment to evaluating our work.
> > >
> > > Best regards,
> > >
> > > The Authors

---

### Decision · Program_Chairs · 2026-04-30

**Decision:**

Reject

**Comment:**

This paper presents ExpAct, a training-free experience layer for mobile GUI agents that stores a widget-level structural graph of interface topology together with subtask-conditioned procedural patterns (including failed trajectories as negative constraints), and reuses them via a three-stage pipeline of abstraction, iterative re-execution, and subtask-wise retrieval. It improves over zero-shot and naive-text-memory baselines on AndroidWorld using UI-TARS-1.5-7B and Qwen2.5-VL-7B, and introduces AndroidWorld-LTB, a 50-task long-tail subset.

Reviewer ratings were positive post-rebuttal (4/5/5 with confidence 4/3/4). The main pre-rebuttal concerns were novelty relative to standard RAG / experience replay (which the authors conceded is an incremental combination) and a critical missing comparison against Mobile-Agent-E, the closest hierarchical-memory method. In rebuttal the authors added the Mobile-Agent-E comparison on AndroidWorld, cross-environment AndroidLab results, and end-to-end latency numbers, which moved reviewer Nza5 from reject to accept.

Despite overall positive scores from the reviewers, I recommend rejection for the following two reasons:
(1) for a paper whose novelty is an integration of known components, the submission benchmarks against only a single memory-augmented competitor (Mobile-Agent-E, in rebuttal only), while other directly comparable approaches are absent, and
(2) the ~55% absolute number sits below current AndroidWorld frontier systems, leaving it unclear whether ExpAct is broadly beneficial or mainly compensates for mid-tier VLM limitations.

I believe a revised submission with more memory-augmented baselines plus a frontier-tier base-agent comparison would make a more compelling case.